# First close insight into global daily gapless 1 km PM$_{2.5}$ pollution, variability, and health impact

Jing Wei [1] ✉, Zhanqing Li [1] ✉, Alexei Lyapustin[2], Jun Wang [3], Oleg Dubovik [4], Joel Schwartz[5], Lin Sun[6], Chi Li [7], Song Liu[8] & Tong Zhu [9]

Here we retrieve global daily 1 km gapless PM$_{2.5}$ concentrations via machine learning and big data, revealing its spatiotemporal variability at an exceptionally detailed level everywhere every day from 2017 to 2022, valuable for air quality monitoring, climate change, and public health studies. We find that 96%, 82%, and 53% of Earth's populated areas are exposed to unhealthy air for at least one day, one week, and one month in 2022, respectively. Strong disparities in exposure risks and duration are exhibited between developed and developing countries, urban and rural areas, and different parts of cities. Wave-like dramatic changes in air quality are clearly seen around the world before, during, and after the COVID-19 lockdowns, as is the mortality burden linked to fluctuating air pollution events. Encouragingly, only approximately one-third of all countries return to pre-pandemic pollution levels. Many nature-induced air pollution episodes are also revealed, such as biomass burning.

Ambient air pollution poses a major global environmental concern, ranked as the fourth highest risk factor for human health, causing ~6.7 million global deaths in 2019 according to the Global Burden of Disease study[1]. Exposure to PM$_{2.5}$ (particulate matter with aerodynamic diameters ≤ 2.5 μm) pollution is associated with a variety of circulatory and respiratory diseases[2–5], contributing to 4 million deaths globally in 2020. PM$_{2.5}$ emitted from different sources may have different health risks, especially ultrafine particles from urban traffic or combustion, including black-carbon-containing particles from wildfire smoke[6–9]. Besides long-term effects, short-term exposure to high levels of pollution is also a focal point in environmental health studies[10–14]. With the accumulating evidence of the health effects of PM$_{2.5}$, the World Health Organization (WHO) updated its short-term and long-term air quality guidelines in 2021, aimed at a continual reduction in air pollution[15].

Despite a steady increase in PM$_{2.5}$ monitoring capabilities over the last decade, many people still live far from monitors. The lack of daily global high-resolution PM$_{2.5}$ data has been hindering our ability to assess air pollution exposure and health effects in all parts of the world.

Regional PM$_{2.5}$ estimates have been extensively studied from the satellite perspective using retrieved aerosol optical depth (AOD) products[16–19], but only a handful of studies have been conducted on the global PM$_{2.5}$ scale. Lary et al.[20] trained an ensemble machine-learning model to estimate daily PM$_{2.5}$ from the Sea-viewing Wide Field-of-view Sensor (SeaWiFS) and Terra and Aqua Moderate Resolution Imaging Spectroradiometer (MODIS) Deep Blue AOD products at a 10 km resolution. van Donkelaar et al.[21] derived global daily PM$_{2.5}$ from MODIS and Multiangle Imaging Spectroradiometer (MISR) AOD products via a vertical conversion factor determined from the 3-D

[1]Department of Atmospheric and Oceanic Science, Earth System Science Interdisciplinary Center, University of Maryland, College Park, MD, USA. [2]Laboratory for Atmospheres, NASA Goddard Space Flight Center, Greenbelt, MD, USA. [3]Department of Chemical and Biochemical Engineering, Iowa Technology Institute, The University of Iowa, Iowa City, IA, USA. [4]Laboratoire d'Optique Atmosphérique, Université de Lille, CNRS, Lille, France. [5]Department of Environmental Health, Harvard TH Chan School of Public Health, Boston, MA, USA. [6]College of Geodesy and Geomatics, Shandong University of Science and Technology, Qingdao, China. [7]Department of Energy, Environmental and Chemical Engineering, Washington University in St. Louis, St. Louis, Missouri, USA. [8]School of Environmental Science and Engineering, Southern University of Science and Technology, Shenzhen, China. [9]State Key Joint Laboratory of Environmental Simulation and Pollution Control, College of Environmental Sciences and Engineering, Peking University, Beijing, China. ✉ e-mail: weijing_rs@163.com; zhanqing@umd.edu

Goddard Earth Observing System (GEOS)-Chem chemical transport model (CTM) at a 10 km resolution. Yu et al.[22] developed a stacked machine-learning model to estimate global daily 10 km $PM_{2.5}$ from 2000 to 2019, integrating GEOS-Chem $PM_{2.5}$ simulations with meteorological and geographical data but omitting AOD. Recently, the spatial resolution of $PM_{2.5}$ has improved to 1 km through the inclusion of the MODIS Multi-Angle Implementation of Atmospheric Correction (MAIAC) AOD product[23], combining the Geographically Weighted Regression (GWR) and CTM models. However, global 1 km $PM_{2.5}$ estimates are only available on annual[24,25] or monthly[26] scales.

Global, daily, high-resolution, and high-quality $PM_{2.5}$ data are particularly in demand for a wide range of studies, from air quality to climate change and public health, but monitoring it from space is extremely challenging. $PM_{2.5}$ can vary dynamically in space and time, especially over land, exhibiting strong spatiotemporal heterogeneities[27] caused by many complex and diverse factors. They are sensitive not only to pollution emission profiles, the density of the human population, and meteorological conditions, but also to regionally variant geographic location and terrain conditions[28]. Observing the heterogeneities requires dense in situ monitoring networks, which only exist in a handful of countries for limited periods. They are very unevenly distributed, dense in developed and populated regions but sparse or missing in under-developed and rural regions, posing great challenges in spatial extrapolation[17]. Space-borne remote sensing offers global and uniform coverage, but it is only feasible under clear-sky conditions. The ubiquitous presence of clouds (60–70% globally) would lead to very large data gaps[17,29], seriously hindering our ability to observe air pollution routinely, exacerbated by inherent difficulties in separating clouds from severe haze and heavy smoke, both of particular concern to the public.

These challenges can be tackled by applying machine learning (ML) models to a variety of datasets. A state-of-the-art ML model allows for the extraction and integration of virtually all information pertaining to $PM_{2.5}$ from ground measurements, satellite remote sensing products, CTM simulations of aerosols and $PM_{2.5}$, as well as population density, topography, atmospheric reanalysis of meteorological fields, pollutant emission inventory, and economic level of regional development. In particular, our ML-extended model takes advantage of model simulations and accounts for the spatiotemporal autocorrelations and differences in air pollutants. It first solves the missing data in satellite aerosol products over cloudy and snow/ice surfaces, then retrieves $PM_{2.5}$ from gap-filled AOD. This leads to spatially complete and more accurate daily $PM_{2.5}$ information for exposure risk and health burden assessments.

Following painstaking efforts in choosing/refining ML models and compiling big datasets from a wide range of sources, we generated a long-term (1 January 2017 to 31 December 2022) global (over land), daily, 1 km resolution, gapless $PM_{2.5}$ dataset for the first time, and used it to address some important questions, such as (1) What are the local, regional, and global $PM_{2.5}$ concentrations on a given day?; (2) What is the daily risk exposure to $PM_{2.5}$ pollution and its associated impact on public health?; (3) How big is the impact of wildfires and COVID-19 episodes on local to global scales?; among others.

## Results and discussion
### Model validation and uncertainty
Our model can accurately estimate daily $PM_{2.5}$ concentrations at most ground-based monitoring stations, as confirmed by high sample-based cross-validated coefficients of determination (CV-$R^2 > 0.6$), low root-mean-square error (RMSE < 10 µg m$^{-3}$), and normalized RMSE scaling by mean (NRMSE < 0.6) values at more than 80% of the stations (Fig. 1a, b). On average, our model achieves a highly reliable overall accuracy, with an average sample-based CV-$R^2$ of 0.91 and RMSE (NRMSE) of 9.2 µg m$^{-3}$ (0.37) for daily retrievals, with accuracies improving on monthly (e.g., CV-$R^2 = 0.97$ and RMSE = 4.15 µg m$^{-3}$) and annual (e.g., CV-$R^2 = 0.98$ and RMSE = 2.77 µg m$^{-3}$) scales (Fig. 1c–e). Additionally,

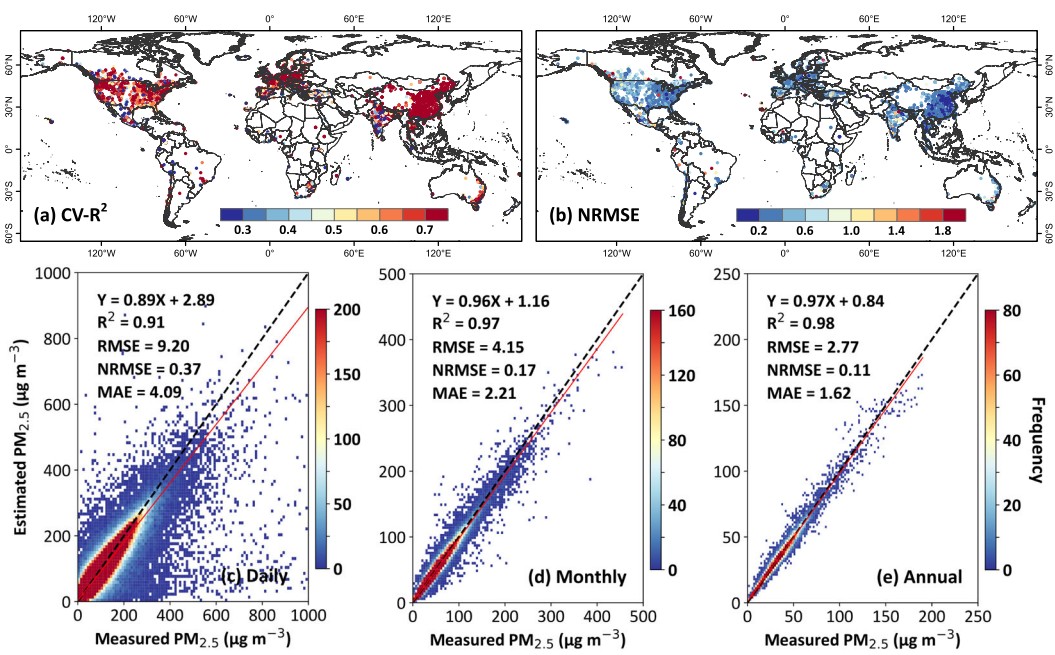

**Fig. 1 | Model validation and uncertainties.** Spatial distributions of sample-based cross-validated (CV) **a** coefficients of determination ($R^2$) and **b** uncertainty (i.e., normalized root-mean-square errors, or NRMSE) of daily $PM_{2.5}$ estimates (unit: µg m$^{-3}$) against ground-based measurements (unit: µg m$^{-3}$) at each monitoring station, and density scatterplots of sample-based CV results between **c** daily estimates (number of samples = 7,089,428), **d** monthly composites (number of samples = 255,075), and **e** annual composites (number of samples = 23,229) and ground-based measurements collected at all monitoring stations from 2017 to 2022 over land. Black dashed lines are 1:1 lines, and red lines are best-fit lines from linear regression. Additional statistical metrics given in **c**–**e** are the linear regression equation, root-mean-square error (RMSE), and mean absolute error (MAE). The maps in **a**, **b** were created using ESRI ArcGIS Pro 3.0.1.

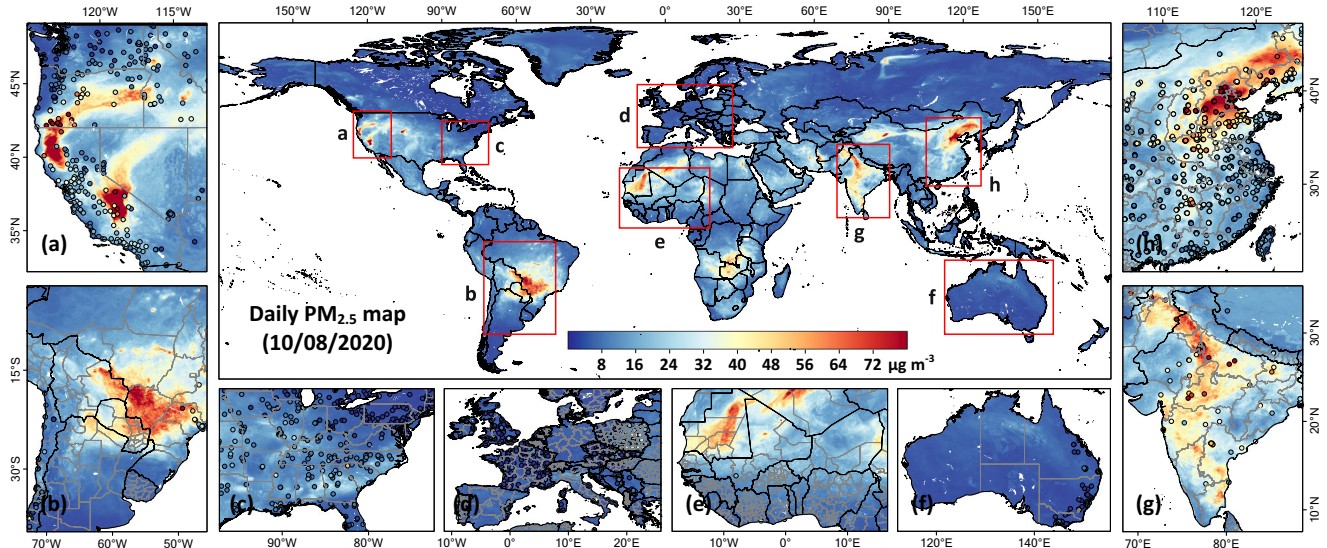

**Fig. 2 | Example of a daily 1 km gapless PM$_{2.5}$ global map on 8 October 2020.** Satellite-derived global PM$_{2.5}$ concentrations (unit: µg m$^{-3}$) at a 1 km spatial resolution and zoomed-in maps (outlined by red rectangles) showing ground-based PM$_{2.5}$ measurements (colored dots) over the **a** western United States, **b** central South America, **c** eastern United States, **d** Europe, **e** northwestern Africa, **f** Australia, **g** India, and **h** eastern China. Thin black lines represent country boundaries or shorelines, and gray lines represent state or provincial boundaries. The maps were created using ESRI ArcGIS Pro 3.0.1.

independent spatiotemporal cross-validation results highlight the model's capability in accurately predicting daily PM$_{2.5}$ levels at locations (i.e., station-, grid-, and state-based CV-$R^2$ = 0.87, 0.79, and 0.76, and RMSE = 10.94, 14.17, and 16.03 µg m$^{-3}$, respectively) as well as on dates lacking ground measurements (i.e., day-, week-, and month-based CV-$R^2$ = 0.81, 0.73, and 0.71, and RMSE = 13.49, 15.01, and 16.46 µg m$^{-3}$, respectively) (Supplementary Fig. S1). Moreover, our model exhibits a comparable "within" $R^2$ of 0.82 to the (station-based) space CV-$R^2$ (0.87), indicating its ability to accurately predict local and daily variations in PM$_{2.5}$ pollution, going beyond capturing only the differences in average values across locations or between seasons. Also, the model's strong performance in continent-stratified CV (cluster CV-$R^2$ = 0.54–0.89) further underscores its robustness in handling large areas with limited training stations (Supplementary Table S1).

**Day-to-day gapless PM$_{2.5}$ variations**

To portray global variations, Fig. 2 presents our 1 km gapless PM$_{2.5}$ retrievals on an individual day (8 October 2020). In general, the spatial patterns of satellite-derived PM$_{2.5}$ retrieval were highly consistent with ground measurements both globally (sample-based CV-$R^2$ = 0.92) and regionally (Fig. 2a–h). Air quality was very good (PM$_{2.5}$ < 8 µg m$^{-3}$) in Canada, the eastern United States (US), Europe, and Australia (Fig. 2c, d, f). By contrast, severe PM$_{2.5}$ pollution occurred in the Beijing-Tianjin-Hebei region in eastern China, the Indo-Gangetic Plain in northern India (Fig. 2g, h), and Zambia in southern Africa, especially around their respective metropolises (i.e., Beijing, New Delhi, and Lusaka), strongly linked with their high population densities and intensive human activities leading to large anthropogenic emissions. Extremely high PM$_{2.5}$ levels (> 80 µg m$^{-3}$) were also caused by smoke from both natural and human-induced fires such as those in the western US (especially in California) and in central South America (Fig. 2a, b). Additionally, high PM$_{2.5}$ concentrations were detected in the Sahara in northern Africa (Fig. 2e) and the Taklamakan Desert in northwestern China, both associated with dust storms. Conventional satellite-based remote sensing approaches are only applicable to clear-sky pixels, resulting in very spotty distributions of PM$_{2.5}$, as illustrated in Supplementary Fig. S2 in contrast to the smooth image of Fig. 2, significantly impeding the ability to discern spatial patterns of PM$_{2.5}$

pollution, especially on small scales, and markedly increasing the probability of missing crucial high-pollution events.

Our gap-filling method[27] allows us to capture PM$_{2.5}$ pollution much more thoroughly to observe strong region-to-region and day-to-day variations throughout the year around the world. While the retrieved PM$_{2.5}$ may be misleading in pointing out the sources of pollution because pollution is inherently both situ and transboundary, they still portray the severity and variations of air pollution problems at various scales, ranging from local and regional to national and continental levels. We can, e.g., clearly see the disparity between developed and developing worlds (categorized by the World Population Review) (Fig. 3a–c) with much lower and higher PM$_{2.5}$ loads, respectively, approximately half (48%). The influences of natural variables may also be inferred: for instance, a shallow planetary boundary layer likely plays a major role in ubiquitous pollution maxima during northern winters[30]. Daily data are invaluable in accurately pinpointing the dates and locations of extreme high-pollution events from natural disasters, such as the wildfires/bushfires that have devastated the United States (US), Brazil, and Australia during the fire seasons, as well as spring dust storms in Nigeria (outlined by red dashed ellipses in Fig. 3d–g). Notably, they provide a clear picture of the spatial dynamics of smoke emissions at various stages of wildfire, encompassing the initial ignition, spreading, intensification, and final suppression, as demonstrated by two examples from the western US (Supplementary Fig. S3) and eastern Australia (Supplementary Fig. S4). Additionally, these data excel in identifying the timing of high-pollution events caused by anthropogenic emissions, as observed in China and India (Fig. 3h, i). They are particularly useful for the continuous monitoring of regional severe haze episodes over time and space, offering insights from the formation to the end, as evident in two examples from eastern China (Supplementary Fig. S5) and northern Asia (Supplementary Fig. S6).

Last, we employed Explainable Machine Learning (XAI) to interpret the driving factors behind daily PM$_{2.5}$ variations by calculating the permutation importance for each feature (Supplementary Note 1 and Fig. S7). Our findings revealed that satellite AOD and modeled PM$_{2.5}$ as the primary global contributors, accounting for 51%, followed by meteorological variables (especially relative humidity and planetary boundary-layer height) ranging from 20% to 73%, and aerosol

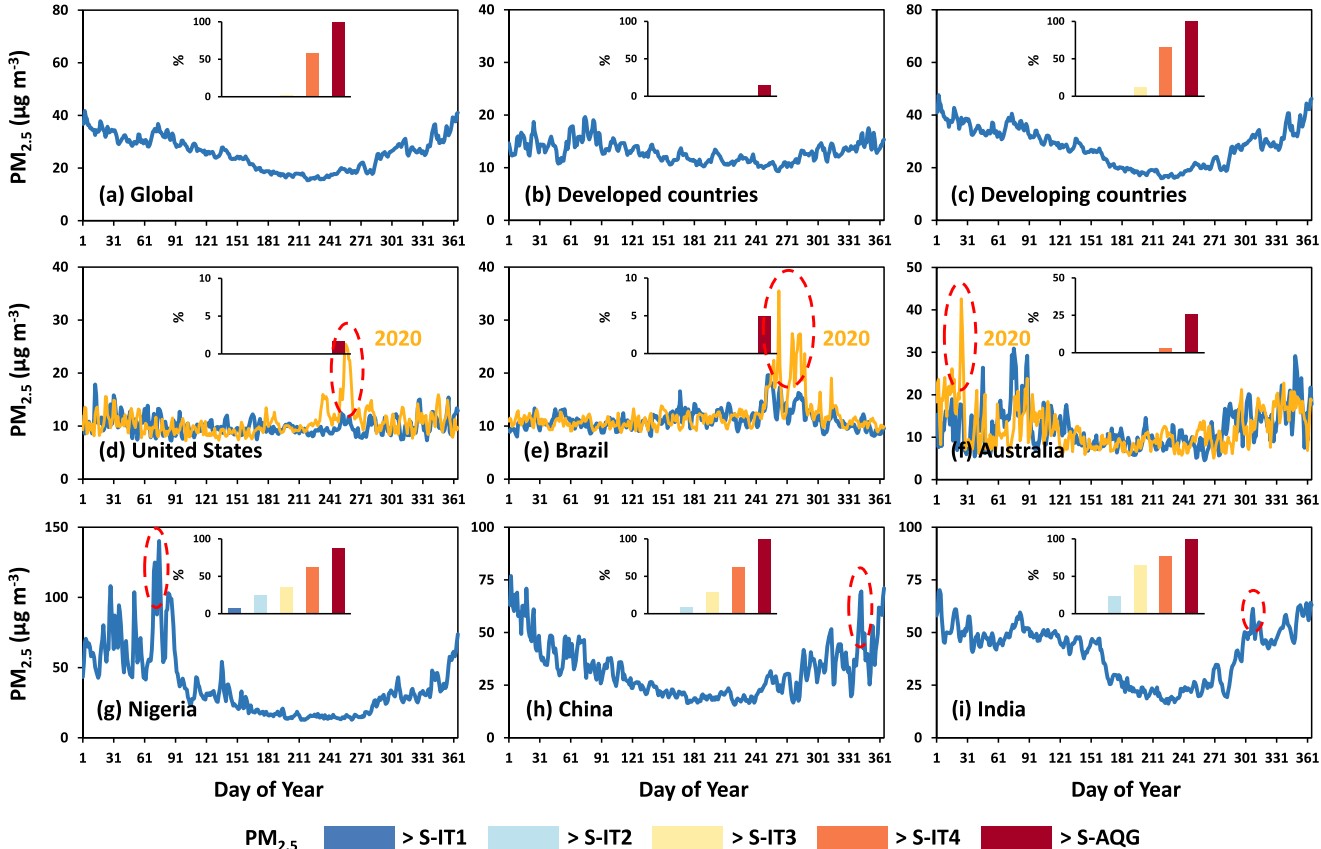

**Fig. 3 | Time series of daily gapless PM$_{2.5}$ for the year 2022.** Time series of daily population-weighted mean PM$_{2.5}$ concentrations (unit: μg m$^{-3}$) as a function of the day of the year in 2022 (blue lines) for the **a** global world, **b** developed, and **c** developing countries, and six selected countries: **d** United States, **e** Brazil, **f** Australia, **g** Nigeria, **h** China, and **i** India. Orange lines in **d**–**f** show the daily PM$_{2.5}$ time series for the El Niño fire year 2020. Red dashed ellipses outline days with anomalously heavy pollution. Inserted bar charts show the percentages of days (unit: %) exceeding the WHO-recommended short-term four interim targets [S-IT1(daily PM$_{2.5}$ > 75 μg m$^{-3}$), S-IT2 (daily PM$_{2.5}$ > 50 μg m$^{-3}$), S-IT3 (daily PM$_{2.5}$ > 37.5 μg m$^{-3}$), S-IT4 (daily PM$_{2.5}$ > 25 μg m$^{-3}$)], and air quality guideline (S-AQG) level (daily PM$_{2.5}$ > 15 μg m$^{-3}$), respectively.

hygroscopicity. The roles of contributing factors vary from region to region due to different causes, which may help provide more effective policies to combat air pollution.

## Daily health-risk exposure to PM$_{2.5}$ pollution

In 2022, global daily population-weighted PM$_{2.5}$ generally satisfied the WHO-recommended short-term interim targets 1 and 2 (i.e., S-IT1 and S-IT2: daily PM$_{2.5}$ = 75 and 50 μg m$^{-3}$, respectively) and came close to achieving the interim target 3 (i.e., S-IT3: daily PM$_{2.5}$ = 37.5 μg m$^{-3}$), with only a few exceptions (3%) on certain days. However, a concerning 57% of the days did not meet the interim target 4 (i.e., S-IT4: daily PM$_{2.5}$ = 20 μg m$^{-3}$), and even worse, all days (100%) exceeded the air quality guidance (i.e., S-AQG: daily PM$_{2.5}$ = 15 μg m$^{-3}$) level (Fig. 3a). Daily PM$_{2.5}$ of developed countries achieved all short-term interim targets, but 15% of the days still exceeded the S-AQG level (Fig. 3b). By contrast, in developing countries, all days met the S-IT1 and S-IT2 targets, but 12% and 66% of the days failed to meet the S-IT3 and S-IT4 targets, respectively, and not a single day met the S-AQG level (Fig. 3c). Overall, only a small percentage of countries (13%) exceeded the S-IT1 target at least once. However, as more stringent air quality standards were imposed, the proportions steadily increased, with 28%, 38%, and 66% of countries failing to meet the S-IT2, S-IT3, and S-IT4 targets, respectively (Fig. 4a–d).

The top 20 countries worldwide recording the highest daily PM$_{2.5}$ exposure risk primarily came from North Africa and the Middle East (65%), and South and East Asia (25%), among which Kuwait, Pakistan, India, and China topped the list, with all days (100%) surpassing the

S-AQG level (Fig. 4e and Supplementary Table 2). A staggering 87%, 80%, and 67% of countries experienced unhealthy air for at least 1, 7, and 30 days, respectively. This phenomenon worsened significantly at the city level, with nearly all (~99.7%) of 1860 major cities (defined as urban agglomerations with populations greater than 300 thousand, according to the United Nations World Urbanization Prospects) being exposed to PM$_{2.5}$ risk (within 10 × 10 km$^2$ around city centers) for at least one day, with exposure periods for 7 days and 30 days reaching 97% and 91%, respectively (Fig. 4e). Despite the proportions consistently decreasing with more stringent targets, a considerable number of cities still faced severe air pollution, with daily PM$_{2.5}$ levels exceeding the S-IT1 target for 44% (26% and 12%) of cities for 1 day (7 and 30 days) (Fig. 4a). Among the top 20 cities, 16, including the capitals of four countries, experienced the most frequent exposure risk, with all days (100%) above the S-AQG level, and 10 of them were from Pakistan (Supplementary Table 3).

Figure 5 provides a comprehensive global assessment of the population PM$_{2.5}$ daily exposure risk for 2022 at a fine 1 km$^2$ grid in terms of WHO's recommended short-term four interim targets and air quality guideline level. Except for densely populated areas like the Indo-Gangetic and North China Plains, most regions showed a relatively small number of days with severe pollution above the two S-IT1 and S-IT2 targets of less than 20% (Fig. 5a, b). The proportions were much smaller in many developed countries like those in North America and Europe. However, with the continuous promotion of interim targets 3 and 4, both the areas covered and their respective exposure risk to moderate PM$_{2.5}$ pollution increased (Fig. 5c, d).

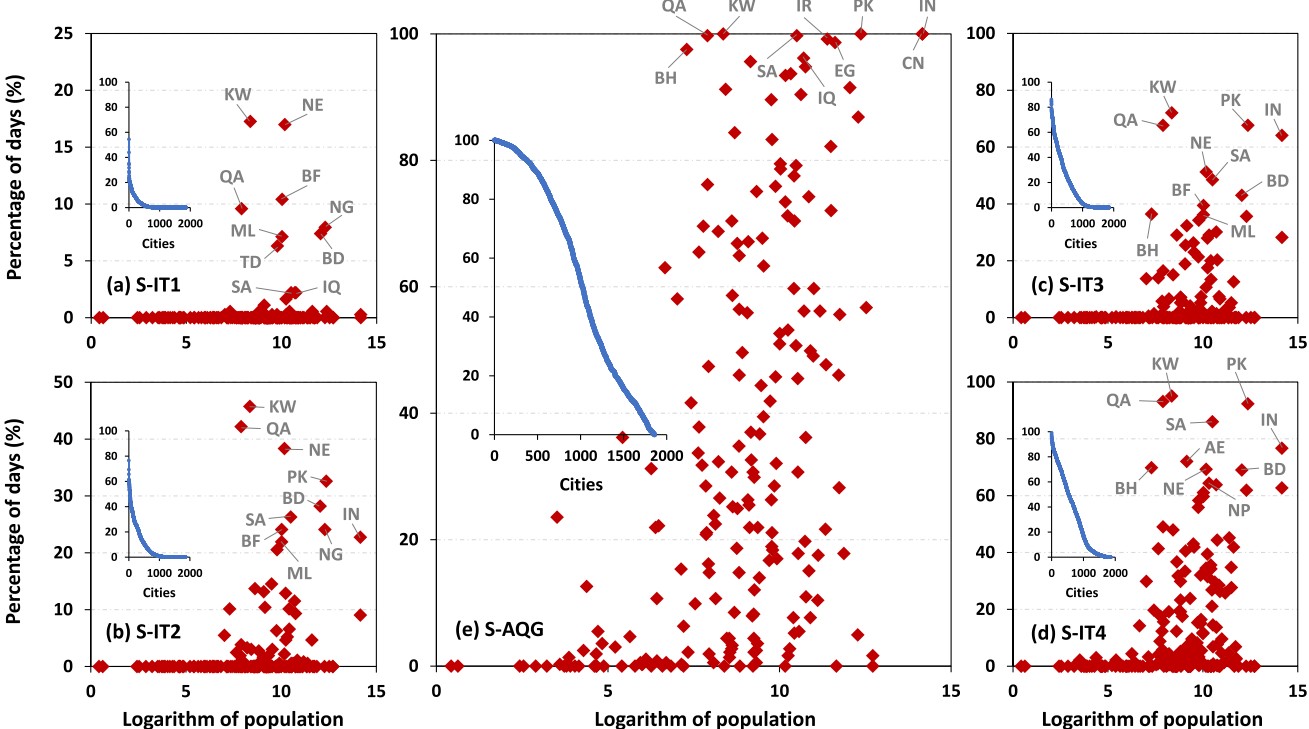

**Fig. 4 | Country- and city-level daily risk exposure to PM$_{2.5}$ pollution.** Scatter distributions of daily exposure risk showing the percentages (unit: %) of days exceeding the WHO-recommended four short-term interim targets: **a** S-IT1 (daily PM$_{2.5}$ > 75 µg m$^{-3}$), **b** S-IT2 (daily PM$_{2.5}$ > 50 µg m$^{-3}$), **c** S-IT3 (daily PM$_{2.5}$ > 37.5 µg m$^{-3}$), **d** S-IT4 (daily PM$_{2.5}$ > 25 µg m$^{-3}$), and **e** short-term air quality guideline (S-AQG) level (daily PM$_{2.5}$ > 15 µg m$^{-3}$) for each country (indicated by red diamonds) as a function of the logarithm of the population (unit: thousand) in 2022. Inserted subplots show the sorted daily exposure risk for each major city (indicated by blue dots). The top ten countries with the highest exposure risks for different WHO daily air quality standards are annotated in gray text. Note that the abbreviations for the labeled countries represent: AE United Arab Emirates, BD Bangladesh, BF Burkina Faso, BH Bahrain, CN China, EG Egypt, IN India, IQ Iraq, IR Iran, KW Kuwait, ML Mali, NE Niger, NG Nigeria, NP Nepal, PK Pakistan, QA Qatar, SA Saudi Arabia, TD Chad.

Regarding the S-AQG level (Fig. 5e), nearly all global inhabited areas were exposed to short-term PM$_{2.5}$ risks, including some developed countries like Italy (A), Poland, and Slovakia, and mega-cities like Los Angeles in the USA (B), with more than 40% of days above the S-AQG level. High daily exposure risks (> 70%) were concentrated in developing countries, particularly those in South Asia. Similar conditions were also observed in many big cities or localities, including Mexico City in Mexico (C), Región Metropolitana de Santiago in Chile (D), Sao Paulo in Brazil (E), Gauteng in South Africa (F), Tehran in Iran (G), Krung Thep in Thailand (H), Ho Chi Minh in Viet Nam (I), and Pyongyang in North Korea (J).

In general, ~23%, 41%, 56%, 80%, and 96% of Earth's populated areas exceeded the WHO S-IT1 through S-IT4 targets and the S-AQG level (Fig. 5i–v), respectively, at least once in 2022. The respective proportions are 13%, 24%, 33%, 52%, and 82% for 7 days, and 5%, 15%, 22%, 32%, and 53% for 30 days in a year. Notably, significant differences existed between developed and developing countries. While the exposure areas for at least 1 day above S-AQG are comparable (96% vs. 96%), the difference rapidly widened with higher exposure risks, e.g., S-IT4 (73% vs. 88%), S-IT3 (39% vs. 75%), S-IT2 (19% vs. 64%), and S-IT1 (6% vs. 41%). More importantly, this gap (e.g., S-AQG) grew significantly by expanding the exposure period to 7 days (74% vs. 90%) and 30 days (29% vs. 78%), emphasizing the greater short-term exposure risk in middle- and low-income counties (Fig. 5vi). Arguably, when the exposure days are related to wildfire or agricultural biomass burning and come in succession rather than randomly during the year, they potentially may lead to more adverse health effects[9,31,32]. These findings also illustrate the global extent of air quality challenges and the need for comprehensive efforts to tackle air pollution and work towards meeting international air quality standards.

## Short-term PM$_{2.5}$ change and mortality burden

El Niño year 2020 stands alone, resulting in large regional contrasting differences: stronger than usual wildfire activity and biomass burning led to a large number of smoke-related PM$_{2.5}$ emissions, resulting in elevated levels of positive PM$_{2.5}$ anomalies in global hot and dry regions during the fire seasons, including the western part of North America[33–35], most areas of South America[36–38], and eastern Australia (Supplementary Fig. S8). Specifically, during the month with the most wildfire records in 2020, population-weighted PM$_{2.5}$ levels were 224%, 80%, and 143% higher in the western US in September, central South America in October, and southeastern Australia in January (outlined in blue in the figure), respectively, compared to historical (2018–2019) levels. Subsequently, we computed the difference in cumulative premature deaths attributed to short-term PM$_{2.5}$ exposure between fire and normal years. Enhanced wildfire smoke-related PM$_{2.5}$ emissions in the western US (September), Brazil (October), and southeastern Australia (January) led to additional deaths of 832 (95% confidence level, or CI: 567, 1122), 449 (95% CI: 305, 592), and 93 (95% CI: 63, 122) people, respectively. However, the above-mentioned seasonal effects, limited in their geographical extent, are superimposed on the longer-term near-global-scale changes brought about by the reduced pollution associated with the COVID-19 pandemic.

To quantify the short-term impact of the COVID-19 lockdown on air quality, we first examined the time series of daily PM$_{2.5}$ variations in the pandemic year 2020, and preceding (2018–2019) and subsequent (2021–2022) baseline years as a function of the Oxford Coronavirus Government Response Tracker (OxCGRT) Stringency Index (SI) in two populous countries (Supplementary Fig. S9). PM$_{2.5}$ pollution responded swiftly to epidemic containment measures: when the SI experienced sharp rises, PM$_{2.5}$ dropped rapidly, whereas when restrictions eased,

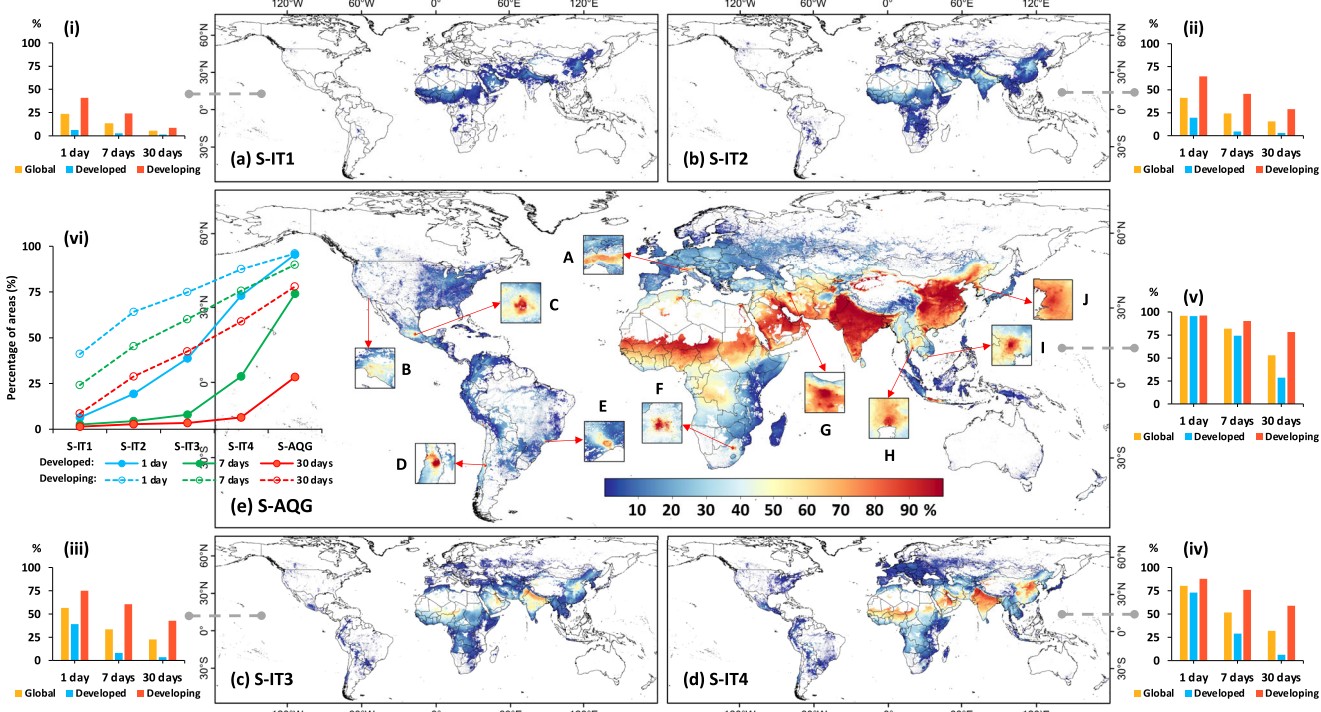

**Fig. 5 | Global daily risk exposure to ambient PM2.5 pollution.** Spatial distributions of global daily exposure risks with the percentages (unit: %) of days exceeding the WHO-recommended four short-term interim targets: **a** S-IT1 (daily $PM_{2.5} > 75\ \mu g\ m^{-3}$), **b** S-IT2 (daily $PM_{2.5} > 50\ \mu g\ m^{-3}$), **c** S-IT3 (daily $PM_{2.5} > 37.5\ \mu g\ m^{-3}$), **d** S-IT4 (daily $PM_{2.5} > 25\ \mu g\ m^{-3}$), and **e** short-term air quality guideline (S-AQG) level (daily $PM_{2.5} > 15\ \mu g\ m^{-3}$) at each 1-km² grid in areas with a population density > 1 per km² for the year 2022. Square insets zoom in on hot spots with high daily PM2.5 exposure risks in mega-cities (A–J). The (i–v) bar charts show the percentages (unit: %) of populated areas that have at least 1, 7, or 30 days exceeding the WHO daily air quality standards for the global, developed, and developing world. The line chart plots in (vi) show the percentages (unit: %) of populated areas as a function of WHO daily air quality standards for developed and developing countries (i.e., solid and dashed lines, respectively) and for exposure periods of 1 day, 7 days, and 30 days (different colored lines). The maps were created using ESRI ArcGIS Pro 3.0.1.

PM2.5 recovered gradually. Remarkably, during the strictest control periods, significant improvements in air quality were observed in China (4 February to 6 March 2020: SI = 79–82)[39] and India (1 March to 12 April 2020: SI = 87–100)[40] compared to the pre-pandemic era, especially in their highly populated provinces (e.g., Shandong and Henan) or states (e.g., Uttar Pradesh and Madhya Pradesh). The average declines in population-weighted PM2.5 concentrations were 14% and 21%, saving 3111 (95% CI: 2132, 4066) and 7667 (95% CI: 5251, 10,027) lives, respectively. Interestingly, during the post-pandemic era, a striking spatially contrasting pattern emerged as PM2.5 rebounded strongly across the entirety of India, with a notable increase of 27%, resulting in an additional loss of 6653 (95% CI: 4557, 8700) lives. By contrast, in China, PM2.5 did not fully recover and even remained about 4% lower than in 2020, saving 450 (95% CI: 310, 586) lives. This disparity can be primarily attributed to China's rigid determination to prevent the epidemic under the 'zero-COVID' policy (especially after the Omicron outbreak in late 2021)[41], as well as the persistent efforts to reduce pollutant emissions[42].

A deeper understanding has been attempted to tackle the following important scientific and social questions. What was the global-scale impact of the COVID-19 lockdown on air quality, and did PM2.5 experience a rebound during the post-pandemic era? Also, what were the benefits or losses of the COVID-19 epidemic to public health? To address these questions, we calculated the changes in PM2.5 and attributed premature deaths during the most stringent lockdown periods both before and after the pandemic for each respective country (Fig. 6). Across the globe, most countries had enacted strict measures to counter epidemics (average lockdown duration = 44 days), e.g., ~94% of them recorded a maximum SI surpassing 60, with an average value of 83. This manifested as negative PM2.5 anomalies seen in ~80% of global

countries, with larger reductions (> 30%) observed in countries in South Asia, northern Europe, and North Africa (Fig. 6a), primarily attributed to a significant decrease in emissions of major pollutants[43], aligning with findings reported in previous studies[44–47]. By contrast, opposite growth trends were observed in only a handful of coastal countries in southern South America, Southeast Asia, and southern Europe[48]. Overall, the global population-weighted PM2.5 in 2020 decreased by ~9% during the lockdown period relative to the pre-pandemic years. The improved air quality resulting from COVID-19 lockdowns yielded significant health benefits for the majority of countries worldwide (Fig. 6b), especially those with dense populations. This led to a notable reduction in the number of premature deaths attributed to short-term PM2.5 exposure, amounting to approximately 19,031 (95% CI: 13,020, 24,915) people on a global scale. Nevertheless, COVID-19 itself was responsible for ~3.3 million deaths in 2020[49].

In the post-pandemic era, noteworthy disparities emerged in the global change in PM2.5 pollution and the associated mortality burden. Substantial increases in both PM2.5 levels and associated premature deaths were observed across most countries in North America, Europe, North Africa, the Middle East, and South Asia (Fig. 6c, d), primarily attributed to a rapid surge in anthropogenic emissions. By contrast, opposite declining trends were found in the majority of countries spanning South America, South Africa, East and Southeast Asia, and Oceania. In general, approximately 59% of countries have undergone a rebound in PM2.5 levels, while the rest have kept below those of 2020. Although these shifts have yielded both favorable and adverse outcomes for air quality and public health, global PM2.5 levels have encountered an approximate 6% increase, leading to an additional global burden of premature deaths, estimated at around 14,444 (95%

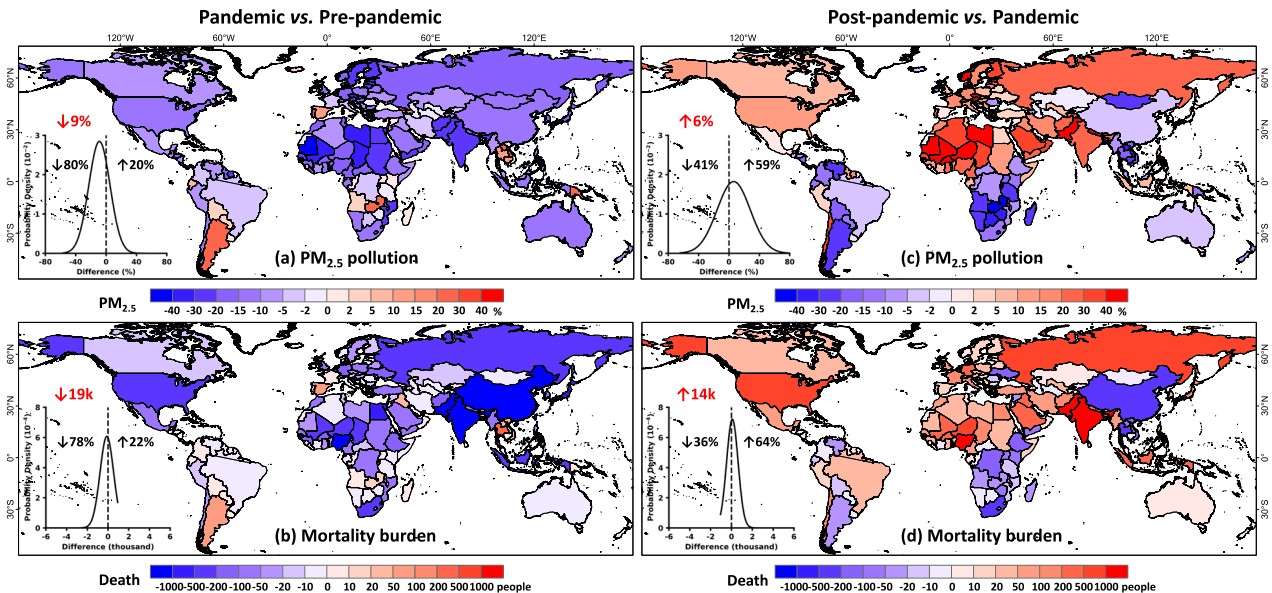

**Fig. 6 | PM$_{2.5}$ change and mortality burden during the COVID-19 lockdown.** Spatial distributions of relative differences (unit: %) in (**a**, **c**) PM$_{2.5}$ pollution and the associated (**b**, **d**) mortality burden (MB, unit: people) for each country during the most stringent period, as determined by the Oxford Coronavirus Government Response Tracker (OxCGRT) stringency index, between the (**a**, **b**) pre-pandemic (2018–2019) and (**c**, **d**) post-pandemic (2021–2022) eras in comparison to the pandemic year (2020). Inserted probability density distributions illustrate changes in PM$_{2.5}$ pollution and associated premature deaths, where the black left- and right-hand numbers indicate percentages of countries with positive (↑) and negative (↓) changes, and red numbers indicate changes in global averages for PM$_{2.5}$ concentrations (unit: %) and total premature deaths (unit: thousand). The maps were created using ESRI ArcGIS Pro 3.0.1.

CI: 9889, 18,896) people. Nevertheless, only 32% of countries have returned to the levels of pollution seen before the pandemic.

### Benefits of 1-km-resolution daily PM$_{2.5}$ data

To evaluate the added value of the 1 km product for investigating air quality and public health, we conducted a sensitivity analysis by aggregating the 1 km data to 10 km. While the results are similar at the global and country levels, substantial differences exist at finer scales (Supplementary Fig. S10). Compared with 10 km data, our 1 km PM$_{2.5}$ data pinpoints the PM$_{2.5}$ pollution and exposure risk at the city level, which concerns the public, as evidenced by a more detailed assessment of pollution inhomogeneity and changes within cities, and the identification of local pollution emissions in populated districts and along transportation routes (brown lines in the figure). This enables us to gain more insights into urban-rural differences, especially in major cities of the world, such as New York City, Sao Paulo, and Johannesburg (the largest cities in the United States, Brazil, and South Africa, respectively), and Rome, Tehran, Seoul, Jakarta, and Guangzhou (the capitals of Italy, Iran, South Korea, Indonesia, and Guangdong Province, China, respectively). In general, the use of 10 km data underestimates daily city-level PM$_{2.5}$ pollution levels (associated with premature mortality) and especially exposure risk, with an average difference of approximately 5% (4%) and 14% relative to the 1 km data. This underestimation is particularly pronounced within core urban regions, which exhibit larger spatial heterogeneities compared to the suburbs, attributed to the loss of spatial details at coarser resolutions, aligning with the findings of previous studies[50–52]. These results highlight the importance of finer spatial-resolution data for targeted air quality monitoring and health assessment, particularly within urban or suburban scales.

### Strengths and Summary

The first global, daily, gapless 1 km PM$_{2.5}$ dataset has been generated under the veil of big data by incorporating ample input data, such as worldwide ground-based observations related to the surface and population, satellite remote sensing products, meteorological

reanalysis, and emission inventories. In particular, MODIS MAIAC satellite AOD retrievals and GEOS Forward Processing (GEOS-FP) AOD simulations are used together to fill the spatial gaps due to the presence of clouds, increasing the data availability by 36%. We have developed a high-performance and explainable ensemble-learning model, which solves the spatiotemporal heterogeneities of air pollutants, as confirmed by various independent spatial and temporal cross-validation approaches. The resulting GlobalHighPM$_{2.5}$ (i.e., global high-resolution and high-quality PM$_{2.5}$) dataset, offering global 1 km gapless PM$_{2.5}$ distributions and variations on a daily basis, will help investigate the health effects of acute PM$_{2.5}$ exposure, a realm that remains relatively unexplored, particularly at finer urban scales. Compared to the currently available 10 km datasets, the much-improved 1 km grid resolves city districts based on residents' income levels, providing valuable utility for environmental justice studies.

Taking advantage of this dataset, we conducted an in-depth examination of global daily PM$_{2.5}$ fluctuations and driving factors, population-exposure risks, and associated mortality burden in each 1 km² grid. Strong day-to-day PM$_{2.5}$ variations, inclusive of extreme high-pollution episodes originating from natural disasters and anthropogenic emissions, are well captured. The use of XAI reveals that aerosols and meteorological factors contribute 51% and 34%, respectively, to the variability of global daily PM$_{2.5}$ estimates. In 2022, 87% of countries, along with nearly all major cities (~99.7%), experienced unhealthy air quality on at least one day. The global fractions of the Earth's populated area surpassing the WHO-recommended daily PM$_{2.5}$ exposure thresholds of 15, 25, 37.5, 50, and 75 μg m$^{-3}$ were 96%, 80%, 56%, 41%, and 23%, respectively. The discrepancies between developed and developing countries rapidly grow as exposure risk levels rise and exposure periods expand. Achieving the objectives of the WHO air quality guidelines globally thus remains a challenge requiring a joint international effort. Our dataset also proved invaluable in capturing the intricate dynamics of PM$_{2.5}$ pollution and health outcomes resulting from acute events like the global spread of COVID-19. PM$_{2.5}$ pollution in 80% of countries has decreased, presumably in response to the implementation of the strictest lockdown

 

measures, saving approximately 19.0 (95% CI: 13.0, 24.9) thousand lives. However, after the pandemic, 59% of countries have experienced a rebound in PM$_{2.5}$ pollution compared to 2020, resulting in 14.4 (95% CI: 9.9, 18.9) thousand lives lost; merely 32% of countries have reverted to the PM$_{2.5}$ levels experienced prior to the pandemic.

## Methods

### Big data

Hourly PM$_{2.5}$ measurements from 2017 and 2022 were collected at approximately ~9500 ground-based monitoring stations around the world (Supplementary Fig. S11). Sources of data include OpenAQ, the China National Environmental Monitoring Centre, the US Environmental Protection Agency, the Canadian National Air Pollution Surveillance Program, the European Air Quality e-Reporting, and other national networks (e.g., South Africa, New Zealand, and Brazil). A substantial majority (~74%) of the stations have been and continue to be dedicated to collecting long-term observations spanning at least two years. Raw data has undergone further quality control measures, including the removal of outliers, such as negative, repeating (occurring for more than three continuous hours), and extreme (exceeding the 99.9th quantiles) values. Subsequently, days having at least 20% of valid hourly PM$_{2.5}$ measurements were identified and averaged to obtain daily means at each monitoring site[22,53]. To ensure consistency with auxiliary variables, all daily PM$_{2.5}$ measurements were adjusted to the UTC time zone and then used for independent training and validation of ML models.

Satellite AOD plays an increasing role in the global mapping of PM$_{2.5}$ as the number of ground PM$_{2.5}$ monitoring stations decreases[54]. The MAIAC AOD product (MCD19A2) has the highest spatial resolution of 1 km among all MODIS operational aerosol products over land with considerable accuracy[23]. This product was thus selected as the main predictor for estimating PM$_{2.5}$ concentrations. Terra (~10:30 AM overpass time) and Aqua (~13:30 PM overpass time) MAIAC daily 1 km AOD retrievals at 550 nm with recommended quality assurances[53] from 2017 to 2022 were used. Spatially complete AOD assimilations (550 nm) from the GEOS-FP system every 3 hours at a horizontal resolution of 0.25° × 0.3125° were employed to fill satellite AOD gaps. The global AErosol RObotic NETwork (AERONET) network provides instantaneous AOD measurements made at ~500 stations over land every 15 min (Supplementary Fig. S11). Only recommended high-quality (Level 2.0) AOD data[55] at 550 nm interpolated by the quadratic polynomial fitting method[56] were used to validate the satellite gap-filled AODs.

First, hourly PM$_{2.5}$ simulations from the GEOS Composition Forecast (GEOS-CF) system at a horizontal resolution of 0.25° × 0.25° were employed[57]. Considering secondary formation via chemical reactions, four main precursors of PM$_{2.5}$, i.e., ammonia, nitrogen oxides, sulfur dioxide, and volatile organic compounds, are also involved, provided by the Copernicus Atmosphere Monitoring Service (CAMS) global high-resolution (~0.1° × 0.1°, monthly) emission inventory[58]. Meteorological variables and their vertical profiles affect air pollution, including temperature, humidity, wind, pressure, precipitation, evaporation, and boundary-layer height. Therefore, these meteorological data were collected from hourly ERA5-Land (~0.1° × 0.1°) and ERA5 global (~0.25° × 0.25°) reanalysis datasets[59,60]. In addition, population density, economic level, land cover, and terrain changes impacting air pollution were also considered, directly or indirectly represented by highly relevant and available satellite remote sensing products, i.e., global high-resolution annual WorldPop unconstrained population (1 km)[61], monthly Visible Infrared Imaging Radiometer Suite (VIIRS) nighttime lights (500 m)[62], monthly MOD13A3 normalized difference vegetation index (NDVI; 1 km), and Shuttle Radar Topography Mission (SRTM) digital elevation model (DEM; 90 m) products. In total, 19 independent variables (Supplementary Table 4) were included in this study. Here, the higher-spatial-resolution variables were aggregated, while the low-spatial-resolution variables were resampled to uniform

0.01° × 0.01° grids using the bilinear interpolation approach[19] and used for subsequent air pollution modeling.

### Air pollution modeling

Here, a tree-based ensemble-learning extremely randomized trees (extra-trees)[63] was adopted for modeling air pollutants, whose unique advantages include stronger randomness and an anti-interference ability with reference to other similar types of models of superior performance[19,64]. The model performance can be significantly improved if the spatiotemporal heterogeneity of air pollution is considered during modeling[53]. Therefore, for global modeling, we developed a 4-Dimensional Space-Time Extra-Trees (4D-STET) model by introducing Euclidean spherical space and triangular spiral time into the original ML model (Supplementary Note 2) to better describe both the autocorrelations and differences of individual points in spatial locations (e.g., different global hemispheres) and temporal series (e.g., seasonal cycles)[65], as evidenced by the superior model performance compared to the traditional method (Supplementary Table 5).

The 4D-STET framework (Supplementary Fig. S12) includes two steps:

1. For satellite AOD gap filling, Terra and Aqua MAIAC AOD retrievals were first combined via linear regression conversion models to minimize the biases caused by different observation times and to expand the spatial coverage[53]. Available MAIAC AOD retrievals were then used as true values to train the 4D-STET model together with the main GEOS-FP AOD simulations and potentially influencing factors, i.e., spatially continuous meteorological fields (i.e., boundary-layer height, temperature, humidity, wind, and pressure), land cover, and elevation, as well as spatiotemporal terms. Missing satellite retrievals can thus be computed to generate daily gapless AODs (Supplementary Note 3).

   Filling in both satellite scanning gaps and missing values in MAIAC AOD retrievals over cloudy and snow/ice-covered scenes and some heavy pollution episodes (Supplementary Fig. S13) with the developed ML model significantly increased the spatial coverage from 64% to 100%, providing spatially continuous AOD information at each 1 km grid cell on any given day (Supplementary Fig. S14). AOD is well reconstructed not only in clean-air regions like North America and most of Europe but also over highly polluted and cloudy regions, including India and eastern China (outlined by red rectangles in the figure), with a higher fraction of missing retrievals in MAIAC AOD products. Our daily gap-filled AODs agree reasonably well with AERONET ground-based AOD measurements, with an average correlation (R) of 0.73, which is only marginally lower when compared to AOD retrievals without gap-filling ($R = 0.74$). While gap-filling can inherently introduce some uncertainties, it is a potent technique to achieve full AOD coverage. Leveraging the advanced ML model alongside nonlinear interpolation techniques could help minimize this uncertainty. The substantial increase in daily spatial coverage (~36%), accompanied by a nearly three-fold expansion in the training sample size, could arguably outweigh the slight accuracy loss (~1%) in the gap-filled AOD data.

2. For the surface PM$_{2.5}$ estimation, their ground measurements were regarded as targets, and satellite gap-filled AODs, GEOS-CF PM$_{2.5}$ simulations, and CAMS pollutant emissions were regarded as main predictors. Auxiliary variables affecting PM$_{2.5}$ pollution, including all meteorological fields (i.e., boundary-layer height, temperature, humidity, wind, pressure, precipitation, and evaporation), NDVI, nighttime lights, DEM, population, and spatiotemporal terms, were input to the 4D-STET model to establish a robust AOD-PM$_{2.5}$ relationship. Last, the trained model was employed to retrieve daily PM$_{2.5}$ concentrations for each grid using gapless AOD and other auxiliary variables, relying on the reconstructed relationship (Supplementary Note 3).

 

## Model validation approach

Sample-based, station-based, and day-based ten-fold cross-validation (10-CV) methods[66], three widely used techniques, were used to validate the model, i.e., all data samples, ground monitors, and days were each randomly divided into ten folds, of which nine were used for modeling and one for independent validation. This process ran ten times, in turn, to get the final accuracy after averaging. They were used to evaluate the overall accuracy of the model and its predictive ability in areas and on days without any available observations[53]. Considering the unbalanced distribution of observation stations and the spatio-temporal cluster characteristics of $PM_{2.5}$ pollution, we conducted additional cluster-based spatial (including grid-based and state-based) and temporal (including week-based and month-based) CV approaches[22,67] to enhance our evaluation by expanding the space and time intervals. The cluster-based spatial CV followed a procedure similar to the station-based CV but utilized grid cells ($1° × 1°$) and states as geographic units instead. Our dataset comprised ~1930 grid cells and 700 states across the globe, which were then randomly divided into 10 equal folds. Similarly, the cluster-based temporal CV followed a procedure similar to the day-based CV but utilized weeks and months as date units. This enabled us to effectively isolate spatiotemporal autocorrelations among observation stations and thoroughly evaluate the spatiotemporal predictive ability of our model. In the space CV, we also calculated the "within" $R^2$ by including separate intercepts for each station and each year and assessed the model performance in capturing within-location variations over time[68]. Last, concerning the spatial aggregation in global ground stations, we implemented the continent-stratified CV method by stratifying the monitoring sites within each continent (i.e., North America, South America, Africa, Europe, Asia, and Oceania) and conducting separate space CV[22,69] to evaluate the model's performance across diverse site densities.

## Short-term exposure risk assessment

Population-health exposure risks related to daily ambient $PM_{2.5}$ pollution were assessed according to the global air quality guidelines updated by the WHO in 2021[15] over a given location. Specifically, daily exposure risks were calculated by counting the proportion of days in a year with daily $PM_{2.5}$ concentrations exceeding the WHO-recommended short-term air quality guideline (AQG) level (daily $PM_{2.5} = 15\,\mu g\,m^{-3}$) and four interim targets, i.e., IT4 ($25\,\mu g\,m^{-3}$), IT3 ($37.5\,\mu g\,m^{-3}$), IT2 ($50\,\mu g\,m^{-3}$), and IT1 ($75\,\mu g\,m^{-3}$), respectively. Alternatively, for every day, we can evaluate the proportion of the global populated area (using the $1\,km × 1\,km$ gridded global population database) with $PM_{2.5}$ pollution exceeding the recommended daily air quality standards.

## Impact of the COVID-19 lockdown

2020 was an extraordinary year. Besides being an El Niño fire year, COVID-19 broke out at the beginning of the year and rapidly spread across the world[70]. In response, countries implemented diverse policies, and the Oxford Coronavirus Government Response Tracker (OxCGRT) project developed a daily Stringency Index (SI, ranging from 0 to 100, reflecting the strictness of responses) based on various relevant metrics closely tied to human activities for each country (see details in Hale et al., 2021)[71]. This index provides a means to quantify the short-term impact of COVID-19 lockdowns on global air quality. Specifically, we first utilized the SI to identify periods marked by the most stringent measures enacted in each country, defined as the duration showcasing the most significant escalation and decline of the OxCGRT SI, wherein the maximum value surpasses 20. Subsequently, we investigated the improvement and rebound of air quality by calculating the changes in $PM_{2.5}$ concentrations and associated mortality burden during the most stringent lockdown period for the pre-pandemic (2018–2019) and the post-pandemic (2021–2022) eras compared to the pandemic year (2020). To minimize the influence of meteorological conditions or extreme events like wildfires on our assessment, we used the data that had been adjusted for meteorological factors accordingly[72].

## Acute mortality burden estimation

Health effects, as well as health exposure guidelines, are generally characterized as long-term, short-term, or acute. While long-term exposure is generally well characterized by annual $PM_{2.5}$[24], our daily global $PM_{2.5}$ data allows us to investigate the health impact of exposure to short-term $PM_{2.5}$ pollution, a topic attracting more and more public attention. However, unlike long-term studies[1], no unified exposure-response function is available[10–13]. The acute mortality burden attributable to daily $PM_{2.5}$ exposure was thus assessed by employing an exposure-response function determined by a meta-analysis from a recent review study[14], e.g., the relative risk for mortality from all causes is 1.0065 (95% CI: 1.0044–1.0086) with an increase in daily $PM_{2.5}$ concentration per 10 $\mu g/m^3$. The short-term mortality burden attributable to daily $PM_{2.5}$ pollution can be expressed as:

$$MB_{g,d} = \frac{RR_g(C_g) - 1}{RR_{g,d}(C_g)} \times POP_{g,y} \times \frac{BMR_{c,y}}{N}, \tag{1}$$

where $MB_{g,d}$ represents the short-term mortality burden with estimated premature deaths in grid $g$ on day $d$; $RR_{g,d}(C_{g,d})$ represents the relative risk subject to the daily $PM_{2.5}$ exposure level in grid $g$ on day $d$; $POP_{g,y}$ represents the population in grid $g$ in year $y$; $BMR_{i,j,y,c}$ represents the baseline mortality rate for country $c$ in year $y$; and $N$ represents the number of days in the year. The population data is obtained from the 1 km WorldPop product. To mitigate the impact of the model bias, we made additional adjustments by aligning WorldPop's grid-specific population number with the country-specific population number reported by the United Nations.

## Data availability

Source Data and the global gapless high-resolution and high-quality $PM_{2.5}$ (GlobalHighPM$_{2.5}$) dataset generated in this study have been deposited in the Zenodo database [10.5281/zenodo.6449740] and are publicly available. Other data used in this study are provided in the supporting information (Supplementary Note 4).

## Code availability

All analyses and visualizations in this study are facilitated by data and codes, which have been deposited in the Zenodo database [https://doi.org/10.5281/zenodo.6449740]. Other mapping and data processing are conducted using ArcGIS, Microsoft Excel, and Python.

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

## Acknowledgements
This work was supported by the NASA Applied Science Programs 80NSSC21K1980 (Z.L.) and 80NSSC21K0428 (Ju.W.). The MODIS algorithm maintenance program provides support to A.L. The authors greatly thank Lorena Castro from the University of Iowa for collecting and processing the OpenAQ data, Tianshu Xu from the Beijing Normal University for collecting relevant data and calculating the health burden, and Maureen Cribb from the University of Maryland for editing and polishing the paper.

## Author contributions
Z.L. and Ji.W. conceived and designed the study. Ji.W. performed the research and wrote the initial draft of this paper. Z.L., Ji.W., A.L., Ju.W., O.D., J.S., L.S., C.L., and T.Z. reviewed and edited the paper. Ju.W. also provided computing resources and some additional information, including observational data. S.L. assisted in processing the relevant data and calculating the exposure results.

## Competing interests
The authors declare no competing interests.
