## [Peer Review File · Nature Communications]

First close insight into global daily gapless 1 km PM2.5 pollution, variability, and health impactEditorial Note: This manuscript has been previously reviewed at another journal that is not operating a transparent peer review scheme. This document only contains reviewer comments and rebuttal letters for versions considered at *Nature Communications*.

Reviewer #2 (Remarks to the Author):

This article presents the first 1km, global, daily dataset of ground-level PM_{2.5} concentrations, along with an analysis of some of the insights that can be gained from such a dataset. I found the analysis to be well done and rigorous, and the results to be compelling. It represents a substantial contribution to the literature; high resolution estimates of PM_{2.5} will be critical for future policy interventions, with concerns such as environmental justice and unreported emissions increasingly relying on hyperlocal concentration estimates.

The article lacks the kind of succinct description of a scientific discovery or new way of thinking about things that high-impact, general-readership journal articles usually have. I think part of this is due to the fact that the writing could be a little tighter (as one specific example of this general observation, the section "Importance of satellite AOD in ML modelling" could perhaps be deleted completely, as I have never heard a serious argument that satellites are not useful for air quality observation, and therefore it would not likely be of interest to a general readership). However, if the main point of the paper is taken to be "we have created this new dataset and it can be useful to a lot of people", then I think it makes sense to publish it in a high-impact, general readership journal, as scientists from a wide range of disciplines would benefit from knowing about its existence. So, overall, I would lean toward acceptance, with a recommendation of editing for clarity and concision, perhaps focusing a little more on the paper's broadly relevant results (e.g. wildfires, covid, etc) and less on the more esoteric ones (e.g. XAI).

Reviewer #2 (Remarks to the Author):

This article presents the first 1km, global, daily dataset of ground-level PM_{2.5} concentrations, along with an analysis of some of the insights that can be gained from such a dataset. I found the analysis to be well done and rigorous, and the results to be compelling. It represents a substantial contribution to the literature; high resolution estimates of PM_{2.5} will be critical for future policy interventions, with concerns such as environmental justice and unreported emissions increasingly relying on hyperlocal concentration estimates.

The article lacks the kind of succinct description of a scientific discovery or new way of thinking about things that high-impact, general-readership journal articles usually have. I think part of this is due to the fact that the writing could be a little tighter (as one specific example of this general observation, the section “Importance of satellite AOD in ML modelling” could perhaps be deleted completely, as I have never heard a serious argument that satellites are not useful for air quality observation, and therefore it would not likely be of interest to a general readership). However, if the main point of the paper is taken to be “we have created this new dataset and it can be useful to a lot of people”, then I think it makes sense to publish it in a high-impact, general readership journal, as scientists from a wide range of disciplines would benefit from knowing about its existence. So, overall, I would lean toward acceptance, with a recommendation of editing for clarity and concision, perhaps focusing a little more on the paper’s broadly relevant results (e.g. wildfires, covid, etc) and less on the more esoteric ones (e.g. XAI).

Response: Thanks a lot for the very constructive comments and recommendation of publication. Accordingly, we have made many changes to underline the major merits of our findings of broader significance and interest to multiple disciplines, namely, atmosphere, environment, and public health. The abstract has been re-written, and the section “Importance of satellite AOD in ML modeling” has been removed with a reference to our new publication (Tian et al., 2023). The whole sections “Unravelling daily PM_{2.5} driving factors with XAI” and “Explainable Machine Learning (XAI)” with more specialized content have been moved to the supplementary information, and only a concise description remains in the main text. Besides, we have made revisions throughout the manuscript to enhance its clarity and conciseness, in honor of your comment.

Tian, Z., Wei, J. & Li, Z. How Important Is Satellite-Retrieved Aerosol Optical Depth in Deriving Surface PM_{2.5} Using Machine Learning? *Remote Sens.* **15**, 3780 (2023).